# Tissue-Specific Landscape of Metabolic Dysregulation during Ageing

**DOI:** 10.3390/biom11020235

**Published:** 2021-02-07

**Authors:** Fangrong Zhang, Jakob Kerbl-Knapp, Alena Akhmetshina, Melanie Korbelius, Katharina Barbara Kuentzel, Nemanja Vujić, Gerd Hörl, Margret Paar, Dagmar Kratky, Ernst Steyrer, Tobias Madl

**Affiliations:** 1Gottfried Schatz Research Center for Cell Signaling, Metabolism and Ageing, Molecular Biology and Biochemistry, Medical University of Graz, 8010 Graz, Austria; fangrong.zhang@medunigraz.at (F.Z.); jakob.kerbl-knapp@medunigraz.at (J.K.-K.); alena.akhmetshina@medunigraz.at (A.A.); m.korbelius@medunigraz.at (M.K.); katharina.kuentzel@medunigraz.at (K.B.K.); nemanja.vujic@medunigraz.at (N.V.); dagmar.kratky@medunigraz.at (D.K.); ernst.steyrer@medunigraz.at (E.S.); 2Otto-Loewi Research Center, Physiological Chemistry, Medical University of Graz, 8010 Graz, Austria; gerd.hoerl@medunigraz.at (G.H.); margret.paar@medunigraz.at (M.P.); 3BioTechMed-Graz, 8010 Graz, Austria

**Keywords:** ageing, tissue-specific, metabolomics, biomarker

## Abstract

The dysregulation of cellular metabolism is a hallmark of ageing. To understand the metabolic changes that occur as a consequence of the ageing process and to find biomarkers for age-related diseases, we conducted metabolomic analyses of the brain, heart, kidney, liver, lung and spleen in young (9–10 weeks) and old (96–104 weeks) wild-type mice [mixed genetic background of 129/J and C57BL/6] using NMR spectroscopy. We found differences in the metabolic fingerprints of all tissues and distinguished several metabolites to be altered in most tissues, suggesting that they may be universal biomarkers of ageing. In addition, we found distinct tissue-clustered sets of metabolites throughout the organism. The associated metabolic changes may reveal novel therapeutic targets for the treatment of ageing and age-related diseases. Moreover, the identified metabolite biomarkers could provide a sensitive molecular read-out to determine the age of biologic tissues and organs and to validate the effectiveness and potential off-target effects of senolytic drug candidates on both a systemic and tissue-specific level.

## 1. Introduction

Ageing might be defined as the process by which structural and functional changes accumulate in an organism over time. Overall, ageing is characterized by a reduction in the ability to maintain metabolic and functional homeostasis in multiple tissues [1]. This can occur in vastly different compartments within the cell, implying that ageing proceeds as a consequence of the interplay between a multitude of pathways, rather than from a single cause. Altogether, the following nine hallmarks are most frequently proposed to be epiphenomena of ageing: genomic instability, telomere attrition, epigenetic alterations, loss of proteostasis, deregulated nutrient-sensing, mitochondrial dysfunction, stem-cell exhaustion, altered intercellular communication and cellular senescence [1]. In particular, senescence has been suggested to contribute to the course of ageing and age-related diseases [2] through imbalanced cellular function, leading to increased DNA damage, generation of reactive metabolites, oxidative stress and inflammation [3,4,5]. These changes can lead to pathophysiological manifestations like tissue atrophy and nerve loss, both of which are common in ageing tissues. In addition, they are associated with age-related pathologies, such as geriatric syndromes, Parkinson’s and Alzheimer’s disease, diabetes mellitus type 2, and atherosclerosis [6,7,8,9,10,11,12,13]. While these conditions differ greatly in their clinical manifestations, they share a common trait of a dysregulated metabolism [13,14,15,16,17,18]. As an example, blood concentrations of branched-chain amino acids (BCAAs), lipids with low carbon numbers, or sugar metabolites are increased in diabetes mellitus type 2 [16,17], whereas methionine, histidine, lysine and phosphatidylethanolamine are increased in patients suffering from Alzheimer’s disease [18]. Moreover, there is increasing evidence that metabolic changes do not only occur as a consequence of ageing processes, but, vice versa, might be drivers thereof [14].

In each organism, tissues are combined in structural and functional units to form organs. Different organs are integrated and connected by blood and lymph vessels to form a whole organism. The tissue conditions may affect basic vital functions and the health status of the entire organism, and vice versa. With respect to age-related alterations in the metabolome at the tissue level, few studies have been performed so far in mice, and even fewer in humans [19,20,21,22]. Mice are a key tool for ageing research due to their relatively short lifespan, which allows monitoring of the ageing process within a reasonable time frame, and due to the ability to manipulate their genes. Ageing research has so far mostly focused on genetically modified mice that mimic progeroid syndromes, and not on animals that age on their own. Therefore, ageing mice, under normal physiological conditions, are a highly valuable model to investigate the changes in metabolites and metabolic pathways as a consequence of the spontaneous deterioration of homeostatic balance over time. Metabolomics enables the capturing of the entire metabolic state of an organism, allows its temporal resolution at distinct time points during the ageing process, and helps to identify altered pathways and biomarkers during ageing and in disease [23,24]. Today’s biomarkers of ageing mainly include phenotypical read-outs such as frailty or grip-strength [25], as well as a small set of molecular markers that need further evaluation, which provide a more general assessment of the physiological age [26]. These biomarkers are an important tool to describe the physiological changes that occur with age, the process of ageing and the occurrence of age-related diseases.

Here, we aimed to provide a comprehensive set of ageing-related metabolic biomarkers in mouse tissues for the identification of tissue-specific and systemic metabolic changes in an ageing organism. To this end, we employed untargeted nuclear magnetic resonance (NMR) spectroscopy and determined changes in polar metabolites in the brain, heart, kidney, liver, lung and spleen of young (9–10 weeks) and aged (96–104 weeks) wild-type mice (mixed genetic background of 129/J and C57BL/6J). We found alterations in the metabolic phenotypes of all tissues, and identified sets of both tissue-specific and systemic metabolite biomarkers of ageing. We identified the following organ-specific biomarkers: (i) BCAAs, uracil and glutamine in the brain, (ii) leucine, isoleucine, valine and 4-aminobutyrate (GABA) in the heart, (iii) succinate and choline in the kidney, (iv) nicotinamide, glycerol and inosine in the liver, (v) lysine, nicotinamide, aspartate and fumarate in the lung, and (vi) taurine and uridine in the spleen. Uridine changed systemically in most tissues, indicating conserved mechanisms of ageing. Our comprehensive metabolic profiling of the key mouse tissues at different ages provides a robust set of metabolic biomarker candidates to study the mechanisms of metabolic reprograming associated with ageing. A deeper understanding of the underlying processes might not only shed light on the causes of age-related pathologies, but also help to discover novel targets for pharmacological interventions. These biomarker candidates could serve as a read-out of the biological age of tissues, and may be utilized to validate the effectiveness of proposed senolytic therapies. Taken together, comprehensive analyses and utilization of metabolomics provide a useful tool to monitor changes of metabolites during the ageing and degenerative process, and may eventually help to increase health span and, thus, the life quality of the aged population.

## 2. Materials and Methods

### 2.1. Animals and Diets

For all experiments, organs were isolated from young (9–10 weeks) and old (96–104 weeks) female wild-type mice (mixed genetic background of 129/J and C57BL/6J) were used (n = 5). Mice were maintained in a clean, temperature-controlled (22 ± 1 °C) environment with a regular light–dark cycle (12 h/12 h) and unlimited access to a chow diet (Altromin 1324, Altromin Spezialfutter GmbH, Lage, Germany) and water. All experiments were performed in accordance with the European Directive 2010/63/EU and approved by the Austrian Federal Ministry of Education, Science and Research.

### 2.2. NMR Sample Preparation, Data Acquisition and Analysis

Organ samples were snap-frozen in liquid nitrogen and stored at −80 °C until analysis. For the NMR metabolomics analysis, 30–50 mg of each organ was resected. To extract metabolites, 400 µL of ice-cold methanol and 200 µL MilliQ H_2_O were added, and the samples were transferred to a tube containing Precellys beads (1.4 mm zirconium oxide beads, Bertin Technologies, Villeurbanne, France) for homogenization by Precellys24 tissue homogenizer (Bertin Technologies, Montigny-le-Bretonneux, France). After centrifugation at 13,000 rpm for 45 min (4 °C), the supernatant was transferred to a fresh tube, and the samples were lyophilized at <1 Torr, 850 rpm, 25 °C for 10 h in a vacuum-drying chamber (Savant Speedvac SPD210 vacuum concentrator), with an attached cooling trap (Savant RVT450 refrigerated vapor trap) and vacuum pump (VLP120) (Thermo Scientific, Waltham, MA, USA). For the NMR experiments, samples were re-dissolved in 500 µL of NMR buffer (0.08 M Na_2_HPO_4_, 5 mM TSP (3-(trimethylsilyl) propionic acid-2,2,3,3-d_4_ sodium salt), 0.04 (*w*/*v*)% NaN_3_ in D_2_O, pH adjusted to 7.4 with 8 M HCl and 5 M NaOH).

The metabolic-profiling analysis was conducted at 310 K using a 600 MHz Bruker Avance Neo NMR spectrometer equipped with a TXI 600S3 probe head. The Carr–Purcell–Meiboom–Gill (CPMG) pulse sequence was used to acquire ^1^H 1D NMR spectra with a pre-saturation for water suppression (cpmgpr1d, 512 scans, 73728 points in F1, 12019.230 Hz spectral width, 1024 transients, recycle delay 4 s) [27,28]. The ^1^H,^13^C heteronuclear single-quantum correlation (HSQC) spectra were recorded with a recycle delay of 1.0 s, spectral widths of 20.8/83.9 ppm, centered at 3.9/50.0 ppm in ^1^H/^13^C, with 2048 and 256 points, respectively, and 8 scans per increment. NMR spectral data were processed as previously described [29]. Briefly, data were processed in Bruker Topspin version 4.0.2 using one-dimensional exponential window multiplication of the FID, Fourier transformation and phase correction. The NMR data were then imported into Matlab2014b; TSP was used as the internal standard for chemical-shift referencing (set to 0 ppm); regions around the water, TSP and methanol signals were excluded; the NMR spectra were aligned; and a probabilistic quotient normalization was performed. Principal component analysis (PCA), orthogonal partial least squares discriminant analysis (O-PLS-DA) and partial least squares-discriminant analysis (PLS-DA) were performed in Matlab2014b and MetaboAnalyst 4.0 [30], as well as all associated data consistency checks and cross-validation. The statistical significance of the determined differences was validated by the quality assessment statistic Q^2^. This measure provides information about cross-validation and is a qualitative measure of consistency between the predicted and original data with a maximum value of 1. Metabolite identification was carried out using Chenomx NMR Suite 8.4 (Chenomx Inc., Edmonton, AB, Canada) and reference compounds. Quantification of metabolites was carried out by signal integration of normalized spectra. For each metabolite, a representative peak with no overlapping signals was identified, the start and end points of the integration were chosen to revolve around that peak, and the area of the peak was integrated by summing up the value for each point. For visualization of our integration approach, the characteristic peaks of selected metabolites are shown in Appendix A, with the area of integration indicated by the black bars. A univariate statistical analysis was carried out using GraphPad Prism 5.01 (GraphPad Software, La Jolla, CA). Data were represented as mean ± standard deviation (SD). The *p*-values were calculated using a two-tailed Student’s t-test for pairwise comparison of variables, and are only given for metabolites with *p* < 0.05.

## 3. Results

Our goal was to establish both tissue-specific profiles and systemic metabolic signatures of ageing, which could serve as a basis for understanding the overall ageing process. ^1^H-NMR spectroscopy is a powerful technique capable of simultaneous identification and quantification of multiple metabolites in complex biological matrices [31]. To better understand the systemic and tissue-specific ageing process and to identify metabolites influenced by age, we carried out metabolic profiling of the brain, heart, kidney, liver, lung and spleen from young and aged mice using an untargeted NMR spectroscopy approach. Using this approach, we were able to identify NMR signals associated with metabolic differences, and assigned the particular peaks to the respective metabolites using metabolite reference databases [32]. The identified respective biomarker candidates will provide a valuable resource for a variety of applications in ageing research and drug discovery.

Using this method, we first determined the metabolic fingerprints of brain samples in young (9–10 weeks) and aged (96–104 weeks) mice. Neurocognitive ageing is characterized by a reduction in the information-processing time and an impaired long-term memory [33], both of which are related to an imbalance in energy metabolism and redox homeostasis [34].The discriminant clustering between brains from young and old mice shown in the orthogonal-partial least squares-discriminant analysis (O-PLS-DA) plot in Figure 1A indicates the underlying differences in the metabolome, supported by the correlation coefficients R²Y up to 0.997 (*p* = 0.02) and a positive Q² of 0.727 (*p* = 0.03), validating the significance of these results. The reduced NMR spectra revealed alterations in the levels of metabolites in mouse brains of different ages (Figure 1B), with decreased concentrations of lactate, methionine, N-acetylaspartate, uridine and inosine. In contrast, concentrations of leucine, isoleucine, valine, glutamine, allantoin, uracil, tyrosine and phenylalanine were increased in the aged mice (Figure 1B,C). The 2D NMR further confirmed the assigned metabolites (Appendix A).

Impaired metabolic flexibility is a hallmark of the ageing heart, with decreased capacity to oxidize fatty acids and increased glucose metabolism [35]. When comparing the metabolic fingerprints between heart samples isolated from young and aged mice, the O-PLS-DA revealed distinct clustering of respective heart samples with correlation coefficients R^2^Y of up to 0.999 (*p* = 0.19) and a Q^2^ of 0.786 (*p* = 0.02) (Figure 2A). Reduced NMR spectra demonstrated altered abundance of metabolites in normalized heart samples (Figure 2B) and indicated decreased uridine concentrations in the hearts of aged mice, whereas the levels of leucine, isoleucine, valine, acetate, GABA, creatine, uracil, tyrosine and phenylalanine were increased (Figure 2C). The 2D HSQC NMR was consistent with the assigned metabolites in Appendix A.

A plethora of abnormalities in kidney structure and function are positively correlated with advancing age [36]. In the kidney, local immune responses induce cellular metabolic reprogramming that changes with ageing [37]. The distinct clustering of kidney samples from young and old mice is shown in the score and validation plots of the O-PLS-DA (Figure 3A). The two clusters show correlation coefficients R^2^Y of up to 0.997 (*p* = 0.03) and Q2 values of 0.898 (*p* = 0.01) (Figure 3A). In the reduced NMR spectra, we found differences in the abundance of 23 age-dependent metabolites (Figure 3B). Leucine, isoleucine, valine, alanine, methionine, glutamate, succinate, aspartate, asparagine, lysine, ethanolamine, choline, glycerol, creatine, serine, uridine, inosine, tyrosine and nicotinamide were decreased in the cohort representing the older mice, whereas the levels of allantoin and uracil were increased (Figure 3C). The 2D HSQC NMR of a representative kidney sample in Appendix A also matched the metabolites assigned based on 1D spectra.

Impaired fatty-acid oxidation and increased de novo lipogenesis in the liver contribute to the risk for age-associated chronic liver disease [38]. Comparable to other organs described above, we also identified two distinct metabolic clusters in livers of old and young mice with correlation coefficients R²Y of up to 0.997 (*p* < 0.01) and a positive Q² of 0.842 (*p* < 0.01) (Figure 4A). The reduced NMR spectra revealed nine metabolites with varying concentrations (Figure 4B). In old mice, the concentrations of lactate, alanine, glycerol, glucose, uridine, inosine, fumarate and nicotinamide were decreased, whereas aspartate was increased (Figure 4C). The assignment of metabolites was confirmed by a 2D HSQC spectra of one liver sample in Appendix A.

Lung ageing is related to structural remodeling, decreased respiratory function and chronic lung diseases, which are closely linked to the ageing process of the immune system [39]. The hierarchical O-PLS-DA score plots (Figure 5A) allowed a clear discrimination between lung samples from young and old mice with correlation coefficients R²Y of up to 0.986 (*p* = 0.26) and a positive Q² of 0.755 (*p* = 0.02). Malonate, fumarate, and nicotinamide were decreased in the old mice, whereas leucine, isoleucine, valine, threonine, methionine, aspartate, lysine, allantoin, tyrosine, and phenylalanine concentrations were increased (Figure 5B,C). Additional confirmation of assigned metabolites of lung sample was provided by a 2D HSQC spectrum shown in Appendix A.

The spleen plays an important role in the immune system. In aged mice, structural changes in the spleen result in a less effective or decreased immune response [40]. O-PLS-DA models clearly discriminated NMR spectra of spleen samples from young and aged mice (Figure 6A). The reduced NMR spectra revealed decreased concentrations of alanine, methionine, glutamate, aspartate, asparagine, lysine, o-phosphocholine, taurine, glycine, uridine, fumarate, tyrosine, and phenylalanine in the aged mice, whereas lactate, glucose, and allantoin concentrations were increased (Figure 6B,C). The 2D HSQC spectrum assignment of a representative spleen sample was found to be consistent with the metabolites assigned based on 1D spectra (Appendix A).

Finally, we compared the abundance of all metabolites between the organs of young and aged mice. Figure 7 depicts a correlation heat map of metabolites in the six investigated tissues from both young and old mice, which exhibited the different distribution patterns of metabolites between different age and organs.

## 4. Discussion

Ageing is a process that gradually increases an organism’s vulnerability and affects multiple biological pathways, including metabolism. The health state of tissues plays a key role in ageing or vice versa, as age-associated organ failure, for example, can lead to the death of an organism. Therefore, revealing the consequences of ageing on specific metabolites in distinct tissues is essential to provide information on the underlying mechanisms, as they explain the metabolic activity in various tissues and provide functional evidence for biochemical activity. By studying metabolic reprogramming in ageing mice using untargeted NMR-based metabolomics, we identified a set of robust biomarkers for ageing in several murine tissues. NMR spectroscopy is a powerful tool in this regard due to its high reproducibility, and simple analysis and interpretation.

In the course of our study, we analyzed six tissues and identified sets of tissue-specific biomarkers of ageing. In brain tissue, we identified 13 metabolites comprising amino acids and their derivatives, such as tyrosine, phenylalanine and N-acetylaspartate. In addition to amino-acid metabolism, the ageing metabolome of mouse brain is characterized by alterations in the purine and pyrimidine metabolism, with a significant increase in uracil [22]. In line, we observed increased uracil in aged mice. Of note, Larsson et al. reported that the elevated plasma levels of BCAAs (isoleucine, leucine and valine) are associated with Alzheimer’s disease [41]. In line with this observation, we also found increased levels of BCAAs in brain lysates of old mice. This phenomenon may be related to the production of the neurotransmitter glutamate, which is known to be altered in the nervous system during ageing [42]. Additionally, our results for changes in glutamine concentrations, in line with a study that investigated metabolites of the motor cortex of the brain in humans of different age (24–68 years) by ^1^H-NMR, indicate that these metabolites represent stable ageing markers in the brain of both mice and humans [43]. Thus, changes in BCAAs, uracil and glutamine are in accordance with recent studies of brain metabolites [41,44].

With ageing, the heart exhibits alterations in amino-acid and purine metabolism. Here we found a set of 10 metabolites significantly changed in the ageing mouse heart. Downregulation of BCAA catabolism in cardiomyocytes has been previously reported to disrupt autophagy, which in turn may be associated with ageing [45,46]. Elevated BCAA levels can therefore be seen as detrimental, in line with large-scale human cohort studies that investigated heart failure [47] and risks of cardiovascular disease [48]. Thus, increased BCAA levels suggest an increased risk for cardiovascular disease in course of ageing. Similarly, the neurotransmitter GABA has been proposed to interfere with cardiac function and was increased by ageing [49,50]. A direct association with cardiac function has previously been demonstrated [47,49,50], rendering leucine, isoleucine, valine and GABA particularly promising as ageing-heart biomarkers.

In the kidney, we identified profound changes in metabolic profiles, with more than 20 metabolites differing between old and young mice. Most metabolic changes were associated with amino-acid, purine/pyrimidine metabolism, and the tricarboxylic acid (TCA) cycle. Among additional metabolites, changes in the choline status might indicate kidney damage [51]. Choline deficiency has been reported to cause kidney damage in rats due to a decrease in the formation of phospholipids, which in turn causes degeneration of the kidney structure [52]. Thus, the decreased choline status in aged mice points to similar mechanisms [51]. Succinate activates the longevity regulator DAF-16 C in *C. elegans*, which increases stress resistance and may extend lifespan [53]. Decreased succinate levels in old mice suggest a more important role of this metabolite in age-related metabolic adaptations than previously assumed. The importance of glutamate is still investigated, but recent publications indicate a tight connection of expression of glutaminases to cellular senescence [54,55]. We could observe lowered glutamate levels in the kidney and spleen of our old mice, although the levels of glutamine remained unchanged in these organs. Choline, succinate and glutamate might be used in the future as promising biomarkers to determine the health- and age-related status of the kidneys.

Metabolic profiles of mouse livers differed substantially in old compared to young mice, with marked changes in the TCA cycle, as shown by the altered levels of alanine, aspartate and fumarate. Ageing reduces glycerol-3-phosphate acyltransferase activity [56] and glycerol [57] in rats, which is consistent with the finding of decreased glycerol concentrations in old mice. Nicotinamide, a poly ADP-ribose synthetase inhibitor, attenuated ischemia-induced liver injury with potent anti-inflammatory effects [58]. Inosine also has an anti-inflammatory potential and has been shown to be decreased in 24-month-old compared to 1.5-month-old rats [59]. We observed decreased nicotinamide and inosine levels in old mice, which may point to chronic inflammation during ageing [60]. Due to the low variability within each group and the clear distinction between the groups, we propose nicotinamide, glycerol and inosine as potential biomarkers for the age status of liver tissues.

In the lung, a panel of 13 metabolites was identified and linked mostly to amino-acid metabolism and the TCA cycle (alanine, aspartate and glutamine). The concentration of the two aromatic amino acids tyrosine and phenylalanine, all three BCAAs and methionine were increased. To date, few metabolome-wide analyses have been performed on lung tissues, and no data at all are available for heathy lung tissue in the context of ageing. Thus, our results could set the base for further investigations, focusing on the general protein biosynthesis activity and its alterations as a potential cause or consequence of the ageing process. Compared to other organs, lung lysates showed smaller differences in their metabolic profiles between the two respective mouse groups. This may suggest that the change in the metabolic phenotype of the lung is a consequence of metabolic derailing in other organs, which in turn affects the lung metabolome. Nevertheless, the levels of lysine, nicotinamide, aspartate and fumarate differed between the two groups, implying these metabolites as biomarkers for the assessment of ageing in lung tissue. In addition to amino acids and metabolites of purine/pyrimidine metabolism, glucose and lactate levels were changed, indicating derailing of the glucose metabolism with age. Altered levels of nicotinamide and inosine in the lung and liver might be linked to the ability of both molecules in triggering inflammatory responses [58].

In summary, the results obtained for the levels of the BCAAs (leucine, isoleucine and valine) are of special interest, since the catabolism of these amino acids does not take place in hepatic cells, but in non-hepatic cells like neurons, cardiomyocytes or the diaphragm [45]. Usually, the degradation of BCAAs in mice is regulated by an enzyme called protein phosphatase 2Cm (PP2Cm), whose mRNA is particularly highly expressed in the brain and heart, highlighting their primary sites of BCAA catabolism [61]. In the context of diseases, BCAAs have been correlated with cardiac pathology, since the expression of their activator PP2Cm can be influenced by stress, and was therefore decreased in conditions like hypertrophy or heart failure. In vivo studies in zebrafish with deficiency of PP2Cm led to a loss of cardiac contractility and premature death, further pointing out the potential role of a unimpaired BCAA catabolism for cardiac health [61]. Since stress signals such as oxidation or genetic damage increase with age, these stress signals may affect the expression of PP2Cm, and a defect in BCAA catabolism may have adverse health effects, possibly explaining the increase of these metabolites in the aged mice [45]. In detail, the increased levels of the BCAAs (leucine, isoleucine and valine) were detected in the brain, heart and lung, whereas no significant changes were seen in the liver. This broadly fits to the hypothesis that accumulating stress signals may influence BCAA levels. Whether there is a causal or close relationship between derailed BCAA metabolism and ageing cannot be answered by our results. Nevertheless, the role of BCAAs in healthy ageing should be further investigated in the future, as this subset of amino acids represents promising biomarkers for the assessment of healthy ageing. mTOR signaling is active in all tissues particularly involved in cell growth, ageing and metabolism, which is presumably important in tissues with high metabolic rates, such as the liver [62]. Here we found significant changes in glucose and glycerol, which are involved in glycolysis/gluconeogenesis, which in turn are downstream targets of the mTOR signaling pathway.

Sixteen metabolites have been found as biomarker candidates for the ageing spleen. Among them, metabolites emerge as components of metabolic pathways linked to amino acid, glucose and lipid metabolism. A decreased concentration of taurine, a sulfur-containing amino acid that augments the proliferative responses of T-cells, indicates a defect in this process in old mice [63] and points to a reduced potential for detoxification of reactive oxygen species [64]. In line with previous results [65], we observed decreased uridine levels in the ageing spleen, suggesting uridine and taurine as robust spleen-specific biomarker candidates for the ageing process. Accordingly, reduced levels of uridine might be associated with increased cellular senescence in all tissues, as uridine has been shown to affect senescence in human mammary epithelial cells [66]. During ageing, senescent cells reduce production of nucleotides, the essential building blocks of DNA, implying that precursors like inosine, uridine, uracil and nicotinamide might also be found at abnormal levels [66]. Taken together, decreased uridine concentrations in the brain, heart, kidney, liver and spleen of aged mice indicate that uridine may be a general biomarker for ageing.

## 5. Conclusions

Research desperately seeks for molecular markers of ageing [67]. Our straightforward workflow of NMR-based untargeted metabolomics together with the identified metabolite biomarker panels is well suited to study the effect of senolytic drug candidates such as dasatinib, quercetin [68], FOXO4-DRI peptide [69], Bcl-2 family inhibitors [70] and Hsp90 inhibitors [71] to increase overall health and enable healthy ageing. Although experiments with mouse models have already been performed to test the efficacy of certain senolytics in the past, these investigations mostly focused on measuring motor activities, frailty or physical characteristics [26]. The ageing-associated, tissue-specific metabolite biomarkers discovered in the current study should provide a novel molecular read-out for ageing-associated changes of an organism. Our results represent a powerful tool for future drug discovery projects to build a bridge between in vitro and in vivo studies, and to validate the molecular efficacy of investigated therapeutics.

Importantly, besides the power of metabolomics for biomarker discovery and validation, identification of metabolites altered in different states of health and disease can help tracing back the pathway(s) causing metabolic derailing during ageing. Following this approach, and assuming that ageing influences the health status, the identification of altered metabolites during ageing is crucial in classifying metabolic pathways closely linked to key ageing processes such as cellular senescence [72]. Senescent cells are “hypermetabolic,” a condition that could potentially be therapeutically targetable. As previously demonstrated, interventions such as rapamycin treatment and methionine restriction impact important aspects of metabolism and delay cellular senescence to extend cellular lifespan [73,74]. How metabolically targeted drugs can achieve sufficient specificity for senescent over non-senescent cells in vivo to allow successful translation remains an open question. Our study provides a protocol to evaluate these metabolism-targeting drugs in vivo, based on both universal and tissue-specific metabolite alterations that accompany the ageing process. In addition, our approach also depicts a metabolite panel for future in vivo NMR studies in living animals.

In this study, NMR-based untargeted metabolomics was applied to investigate the metabolic profile of key tissues in young and old mice. We revealed that ageing is associated with considerable metabolic alterations specifically in amino acids, neurotransmitters and other small molecules. Our study not only generated a high-quality untargeted analysis of ageing metabolism, but also provided a set of metabolic markers that may be used in further translational studies, such as the development of senolytic compounds. Taken together, our approach brought up a metabolite panel for future in vivo magnetic resonance studies.

## Figures and Tables

**Figure 1 biomolecules-11-00235-f001:**
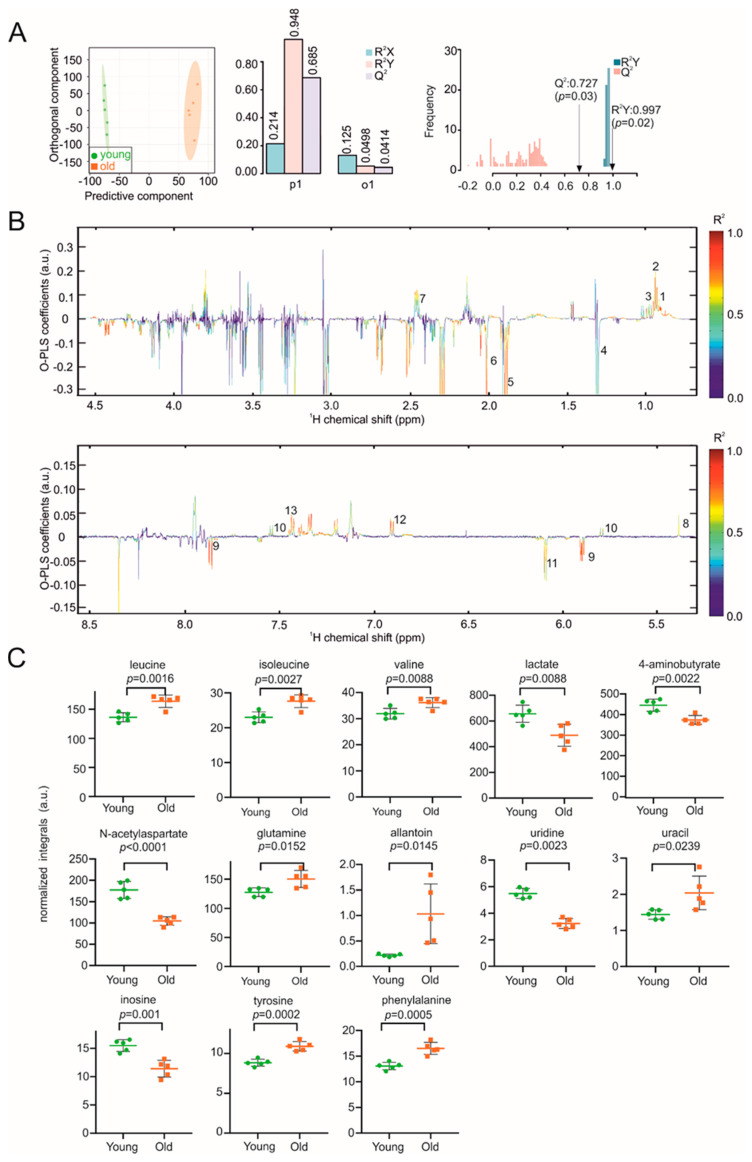
NMR metabolomic analysis of mouse brain samples. (**A**) O-PLS-DA plot of brain samples, including cross validation. (**B**) The reduced NMR spectrum revealed altered components in normalized brain samples. Positive covariance corresponds to components present at increased concentrations, whereas negative covariance corresponds to decreased component concentration. Predictivity of the model is represented by R^2^. 1 = leucine, 2 = isoleucine, 3 = valine, 4 = lactate, 5 = 4-aminobutyrate, 6 = N-acetylaspartate, 7 = glutamine, 8 = allantoin, 9 = uridine, 10 = uracil, 11 = inosine, 12 = tyrosine, 13 = phenylalanine. (**C**) Statistical analysis of altered metabolites in brain samples using a Student’s *t*-test. *p* < 0.05 was considered statistically significant.

**Figure 2 biomolecules-11-00235-f002:**
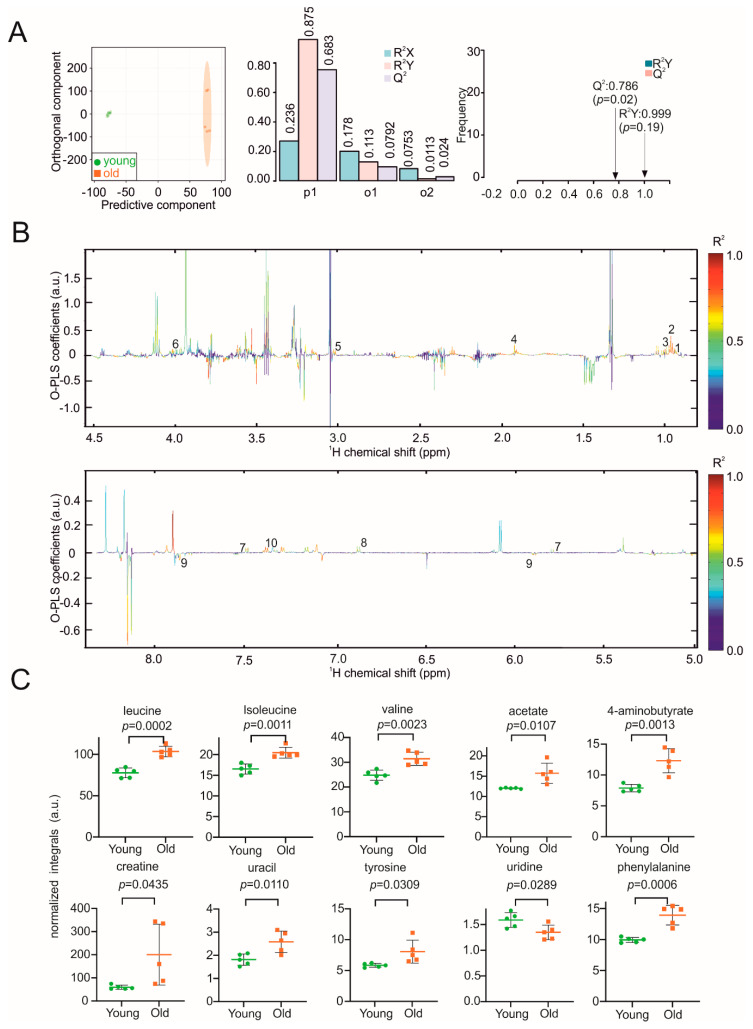
NMR metabolomic analysis of mouse heart samples. (**A**) O-PLS-DA plot of heart samples, including cross validation. (**B**) The reduced NMR spectrum revealed altered components in normalized heart samples. Positive covariance corresponds to components present at increased concentrations, whereas negative covariance corresponds to decreased component concentration. Predictivity of the model is represented by R^2^. 1 = leucine, 2 = isoleucine, 3 = valine, 4 = acetate, 5 = 4-aminobutyrate, 6 = creatine, 7 = uracil, 8 = tyrosine, 9 = uridine, 10 = phenylalanine. (**C**) Statistical analysis of altered metabolites in heart samples using a Student’s *t*-test. *p* < 0.05 was considered statistically significant.

**Figure 3 biomolecules-11-00235-f003:**
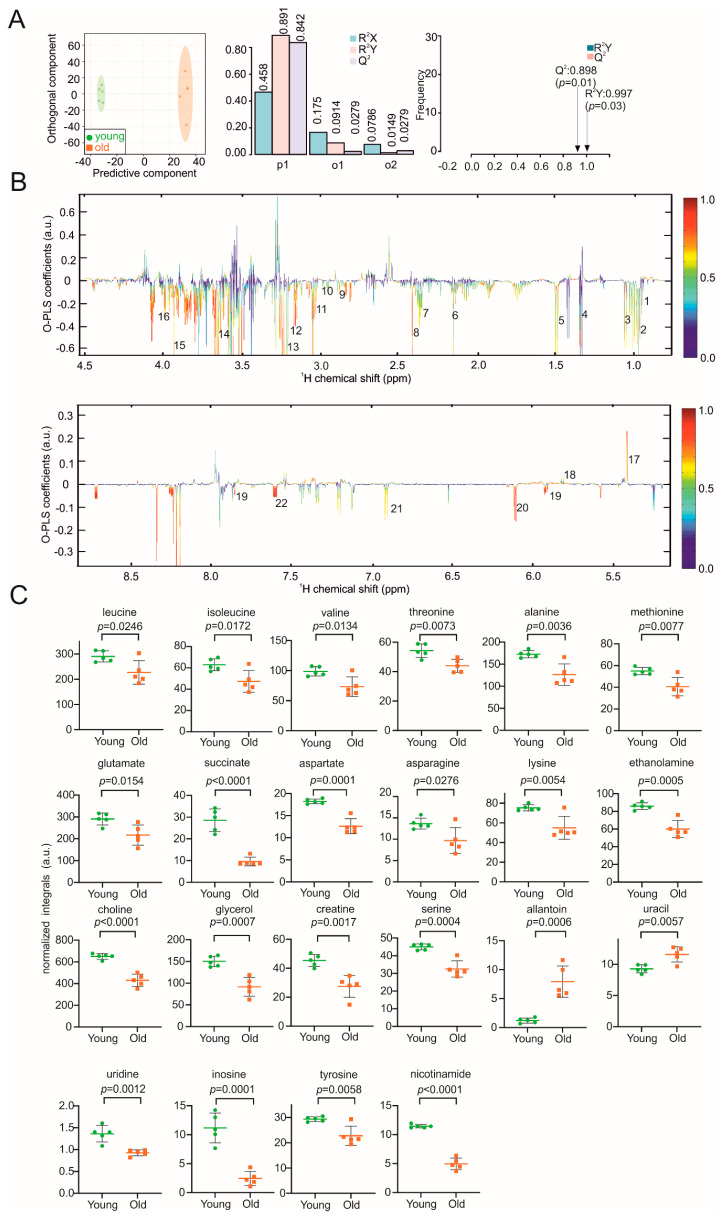
NMR metabolomic analysis of mouse kidney samples. (**A**) O-PLS-DA plot of kidney samples, including cross validation. (**B**) The reduced NMR spectrum revealed altered components in normalized kidney samples. Positive covariance corresponds to components present at increased concentrations, whereas negative covariance corresponds to decreased component concentration. Predictivity of the model is represented by R^2^. 1 = leucine, 2 = isoleucine, 3 = valine, 4 = threonine 5 = alanine, 6 = methionine, 7 = glutamate, 8 = succinate, 9 = aspartate, 10 = asparagine, 11 = lysine, 12 = ethanolamine, 13 = choline, 14 = glycerol, 15 = creatine, 16 = serine, 17 = allantoin, 18 = uracil, 19 = uridine, 20 = inosine, 21 = tyrosine, 22 = nicotinamide. (**C**) Statistical analysis of altered metabolites in kidney samples using a Student’s *t*-test. *p* < 0.05 was considered statistically significant.

**Figure 4 biomolecules-11-00235-f004:**
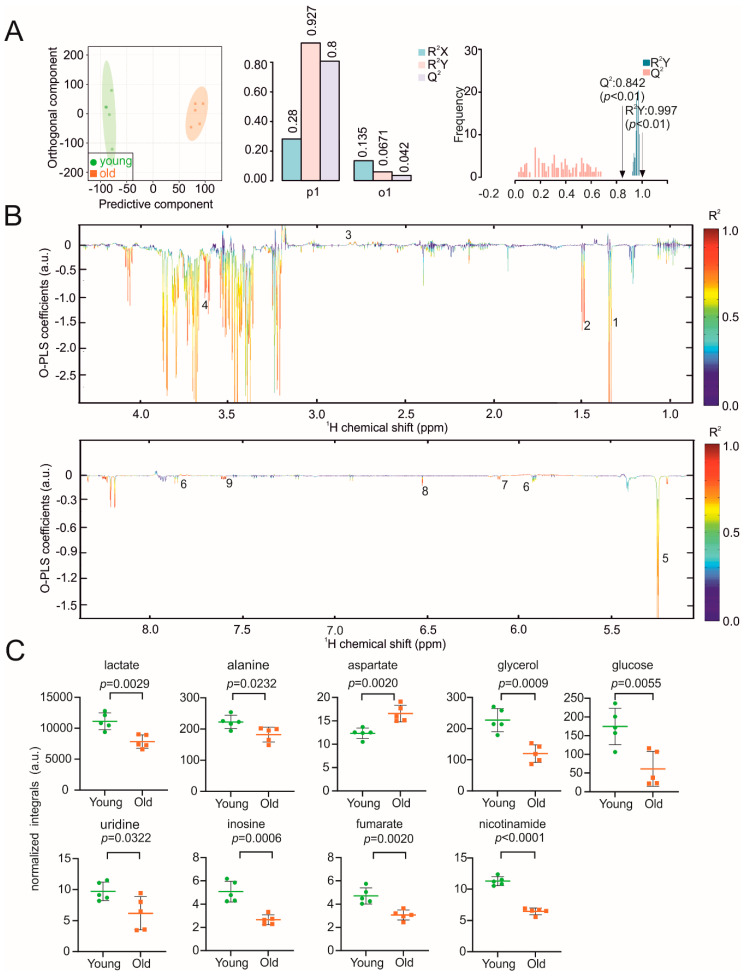
NMR metabolomic analysis of mouse liver samples. (**A**) O-PLS-DA plot of liver samples, including cross validation. (**B**) The reduced NMR spectrum revealed altered components in normalized liver samples. Positive covariance corresponds to components present at increased concentrations, whereas negative covariance corresponds to decreased component concentration. Predictivity of the model is represented by R^2^. 1 = lactate, 2 = alanine, 3 = aspartate, 4 = glycerol, 5 = glucose, 6 = uridine, 7 = inosine, 8 = fumarate, 9 = nicotinamide. (**C**) Statistical analysis of altered metabolites in liver samples using a Student’s *t*-test. *p* < 0.05 was considered statistically significant.

**Figure 5 biomolecules-11-00235-f005:**
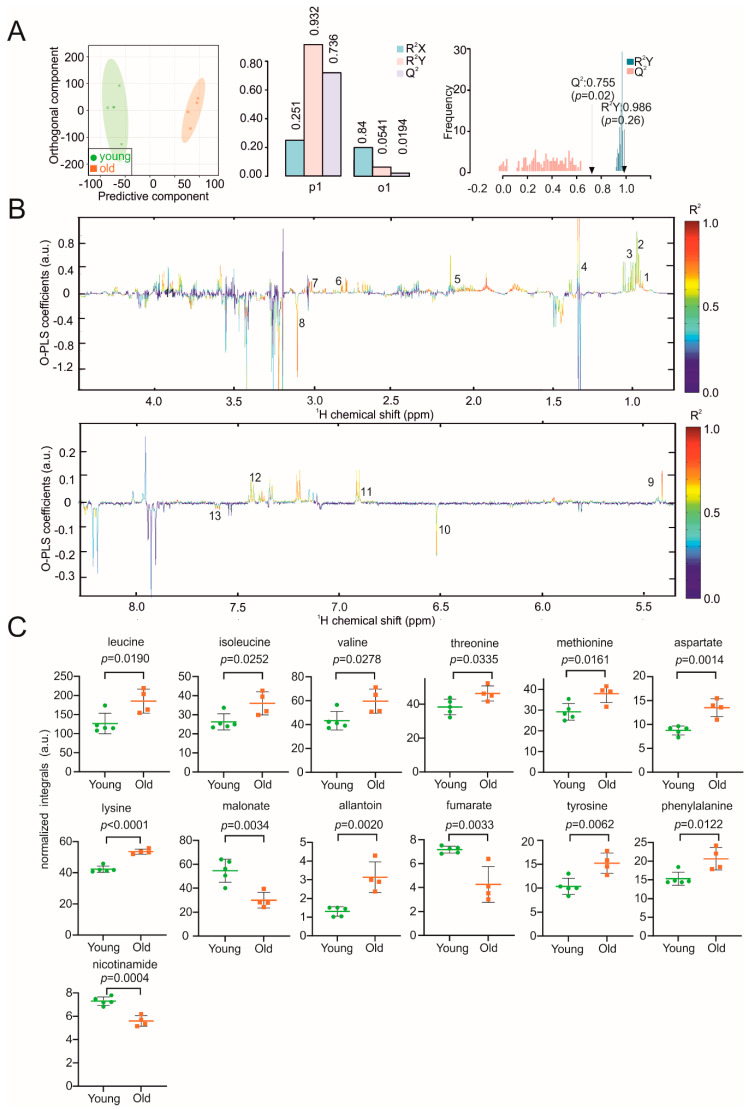
NMR metabolomic analysis of mouse lung samples. (**A**) O-PLS-DA plot of lung samples, including and cross validation. (**B**) The reduced NMR spectrum revealed altered components in normalized lung samples. Positive covariance corresponds to components present at increased concentrations, whereas negative covariance corresponds to decreased component concentration. Predictivity of the model is represented by R^2^. 1 = leucine, 2 = isoleucine, 3 = valine, 4 = threonine, 5 = methionine, 6 = aspartate, 7 = lysine, 8 = malonate, 9 = allantoin, 10 = fumarate, 11 = tyrosine, 12 = phenylalanine, 13 = nicotinamide. (**C**) Statistical analysis of altered metabolites in lung samples using a Student’s *t*-test. *p* < 0.05 was considered statistically significant.

**Figure 6 biomolecules-11-00235-f006:**
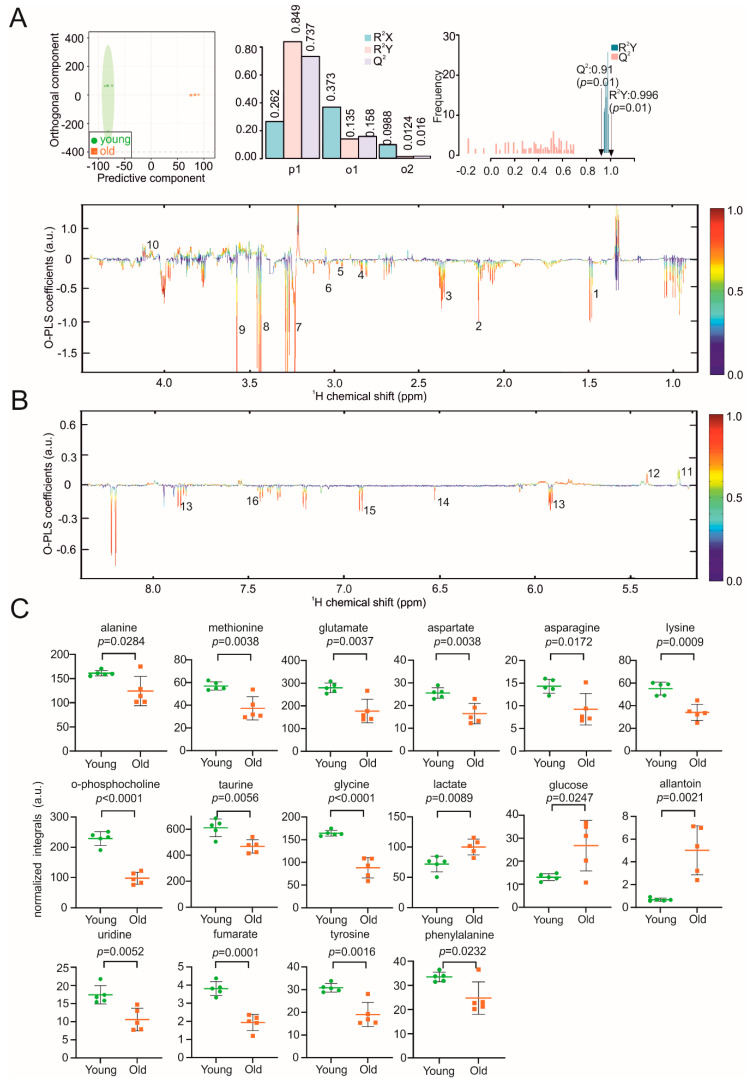
NMR metabolomic analysis of mouse spleen samples. (**A**) O-PLS-DA plot of spleen samples, including cross validation. (**B**) The reduced NMR spectrum revealed altered components in normalized spleen samples. Positive covariance corresponds to components present at increased concentrations, whereas negative covariance corresponds to decreased component concentration. Predictivity of the model is represented by R^2^. 1 = Alanine, 2 = methionine, 3 = glutamate, 4 = aspartate, 5 = asparagine, 6 = lysine, 7 = o-phosphocholine, 8 = taurine, 9 = glycine, 10 = lactate, 11 = glucose, 12 = allantoin, 13 = uridine, 14 = fumarate, 15 = tyrosine, 16 = phenylalanine. (**C**) Statistical analysis of altered metabolites in spleen samples using a Student’s *t*-test. *p* < 0.05 was considered statistically significant.

**Figure 7 biomolecules-11-00235-f007:**
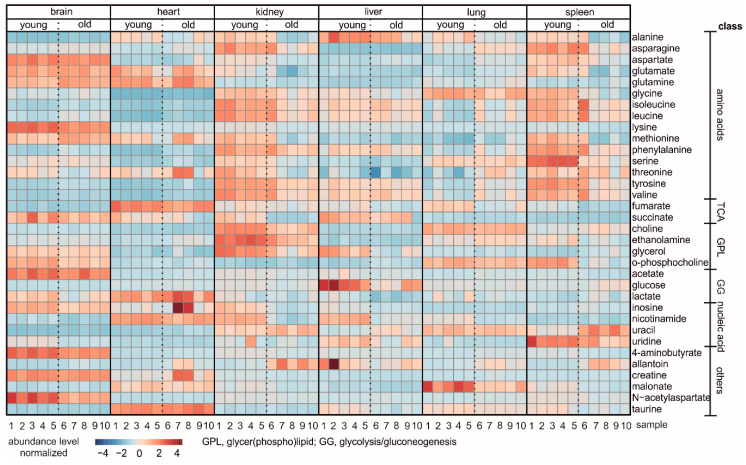
Heat map of NMR analyses showing the relative metabolite levels in organs from young and old mice. Each column represents one single sample, and each row represents one distinct metabolite as indicated. Increased and decreased metabolites are given in red and blue, respectively. Metabolites are indicated and sorted according to different chemical classes or biomolecular pathways. Bioinformatic analysis of data was performed using the statistical package in MetaboAnalyst 5.0.

## Data Availability

The data presented in this study are available on request from the corresponding author.

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
