# Peer review of "Tissue-Specific Landscape of Metabolic Dysregulation during Ageing"

_biomolecules, 2021, doi:10.3390/biom11020235_

Round 1
Reviewer 1 Report
Title: Tissue-Specific Landscape of Metabolic Dysregulation During Ageing
Fangrong Zhang1, Jakob Kerbl-Knapp1, Alena Akhmetshina1, Melanie Korbelius1, Katharina B. Kuentzel1, NemanjaVujić1, Gerd Hörl2, Margret Paar2, Dagmar Kratky1,3, Ernst Steyrer1, Tobias Madl*1,3 and 4
Revisions:
Figures 1-5b should be combined into one summary figure, perhaps with individual metabolites in a heatmap, where the tissues are compared side by side. The minutiae of these three figures can then be put into supplemental data. It does not make sense to have these listed as individual. Once this is done then the authors can use the comparisons to make insightful comments on the analysis and impact on the literature as a whole.
Author Response
Figures 1-5b should be combined into one summary figure, perhaps with individual metabolites in a heatmap, where the tissues are compared side by side. The minutiae of these three figures can then be put into supplemental data. It does not make sense to have these listed as individual. Once this is done then the authors can use the comparisons to make insightful comments on the analysis and impact on the literature as a whole.
We thank reviewer 1 for her/his suggestions. As suggested, we added a heat map summarizing the results and included additional paragraphs in the discussion. We kept panels b in all figures to provide sufficient information for NMR metabolomics researchers and hope this reviewer is ok with this decision.
Reviewer 2 Report
In this work, Zhang et al characterized the tissue-specific metabolic profilings of young and aged mice using NMR, in an effort to identify ageing-related metabolic biomarkers. The work is interesting and convincing. Some minor revisions are recommended.
In details:
1) The authors should provide more details about the lyophilizing process, e.g. instruments, time etc.
2) Please provide citation for the CPMG pulse.
3) line 143, please explain the term 'untargeted NMR spectroscopy'.
4) The line shape fitting and integration of peaks for metabolites with 1D NMR spectroscopy are usually challenging because of the crowding and overlapping of peaks. It is not clearly to me how the authors take care of these issues. The authors might want to provide a little more information on these issues.
5) The authors might also want to zoom into some of the regions and demonstrate the confidence of line shape fitting, spectral deconvolution, peak assignment and integration.
6) Given the caveats mentioned above, the authors might want confirm some of the important metabolic markers with labeled sample and 2D NMR spectroscopy or other methods such as LC-MS.
Author Response
In this work, Zhang et al characterized the tissue-specific metabolic profilings of young and aged mice using NMR, in an effort to identify ageing-related metabolic biomarkers. The work is interesting and convincing. Some minor revisions are recommended.
We thank the reviewer for her/his positive evaluation of our manuscript.
1) The authors should provide more details about the lyophilizing process, e.g. instruments, time etc.
Added
2) Please provide citation for the CPMG pulse.
We have added citations to the pulse sequence.
3) line 143, please explain the term 'untargeted NMR spectroscopy'.
We have added an explanation in the revised version of the manuscript.
4) The line shape fitting and integration of peaks for metabolites with 1D NMR spectroscopy are usually challenging because of the crowding and overlapping of peaks. It is not clearly to me how the authors take care of these issues. The authors might want to provide a little more information on these issues.
We agree with the reviewer that such approaches can be challenging. Here, we used Chenomx for metabolite identification and integrated isolated peaks for univariate analysis shown in panels c of all figures.
5) The authors might also want to zoom into some of the regions and demonstrate the confidence of line shape fitting, spectral deconvolution, peak assignment and integration.
We have added a supplementary figure showing NMR signals integrated for the univariate analysis in the revised version of the manuscript (Figure S1).
6) Given the caveats mentioned above, the authors might want confirm some of the important metabolic markers with labeled sample and 2D NMR spectroscopy or other methods such as LC-MS.
We have included representative natural abundance 1H,13C HSQC spectra for all tissues in the supplement (Figure S2).
Round 2
Reviewer 2 Report
All my concerns are properly addressed in the revised manuscript.